# Identification of AGR2 Gene-Specific Expression Patterns Associated with Epithelial-Mesenchymal Transition

**DOI:** 10.3390/ijms231810845

**Published:** 2022-09-16

**Authors:** Andrea Martisova, Lucia Sommerova, Adam Krejci, Iveta Selingerova, Tamara Kolarova, Filip Zavadil Kokas, Milos Holanek, Jan Podhorec, Tomas Kazda, Roman Hrstka

**Affiliations:** 1Research Centre for Applied Molecular Oncology (RECAMO), Masaryk Memorial Cancer Institute, Zluty Kopec 7, 65653 Brno, Czech Republic; 2National Centre for Biomolecular Research, Faculty of Science, Masaryk University, Kamenice 5, 62500 Brno, Czech Republic; 3Department of Comprehensive Cancer Care, Masaryk Memorial Cancer Institute, Zluty Kopec 7, 65653 Brno, Czech Republic; 4Department of Comprehensive Cancer Care, Faculty of Medicine, Masaryk University, Kamenice 753/5, 62500 Brno, Czech Republic; 5Department of Radiation Oncology, Masaryk Memorial Cancer Institute, Zluty Kopec 7, 65653 Brno, Czech Republic

**Keywords:** AGR2, EMT, TGF-β, RNAseq, arachidonic acid, focal adhesion

## Abstract

The TGF-β signaling pathway is involved in numerous cellular processes, and its deregulation may result in cancer development. One of the key processes in tumor progression and metastasis is epithelial to mesenchymal transition (EMT), in which TGF-β signaling plays important roles. Recently, AGR2 was identified as a crucial component of the cellular machinery responsible for maintaining the epithelial phenotype, thereby interfering with the induction of mesenchymal phenotype cells by TGF-β effects in cancer. Here, we performed transcriptomic profiling of A549 lung cancer cells with CRISPR-Cas9 mediated *AGR2* knockout with and without TGF-β treatment. We identified significant changes in transcripts associated with focal adhesion and eicosanoid production, in particular arachidonic acid metabolism. Changes in transcripts associated with the focal adhesion pathway were validated by RT-qPCR of *COL4A1*, *COL4A2*, *FLNA*, *VAV3*, *VEGFA*, and *VINC* mRNAs. In addition, immunofluorescence showed the formation of stress fibers and vinculin foci in cells without AGR2 and in response to TGF-β treatment, with synergistic effects observed. These findings imply that both AGR2 downregulation and TGF-β have a role in focal adhesion formation and cancer cell migration and invasion. Transcripts associated with arachidonic acid metabolism were downregulated after both *AGR2* knockout and TGF-β treatment and were validated by RT-qPCR of *GPX2*, *PTGS2*, and *PLA2G4A*. Since PGE_2_ is a product of arachidonic acid metabolism, its lowered concentration in media from *AGR2*-knockout cells was confirmed by ELISA. Together, our results demonstrate that AGR2 downregulation and TGF-β have an essential role in focal adhesion formation; moreover, we have identified AGR2 as an important component of the arachidonic acid metabolic pathway.

## 1. Introduction

Progress in human cancer medicine has been driven by a combination of cytogenetic technologies, gene cloning advances, and the use of model organisms, resulting in a more accurate determination of cancer-associated gene functions. In addition, technological advances in transcriptomics and proteomics have significantly improved preclinical research, contributed to the understanding of the complexity of tumors, and elucidated key processes taking place in tumors. The most critical are molecular mechanisms governing the invasion and dissemination of tumor cells, representing an important prerequisite for metastasis development. Although metastatic dissemination is responsible for as much as 90% of cancer-associated mortality, it remains the most poorly understood component of cancer pathogenesis [1]. Metastasis is a multistep process by which tumor cells disseminate from their primary site and form secondary tumors. Initiation of the developmental program termed epithelial-to-mesenchymal transition (EMT) is thought to be a key event in promoting metastasis. This reversible transdifferentiation program is driven by EMT-inducing transcription factors and may induce cancer cells to enter into a stem cell-like state [2]. The execution of EMT in cancer is not homogeneous; it is a spectrum of intermediate states that manifest as a limited focal event at the invasive front of the primary tumor, which is functionally and morphologically distinct from the tumor bulk that remains largely epithelial [3]. Cells at the invasive tumor front with acquired mesenchymal traits communicate with adjacent tumor tissues and the tumor stroma, resulting in their dissociation from the primary tumor and intravasation into blood or lymph vessels [4]. The role of EMT as a driver of cancer progression is also supported by a study in which intravital microscopy detected a pool of breast tumor cells that spontaneously undergo EMT, become motile and disseminate, and then reverse to the epithelial state upon metastatic outgrowth [4].

In recent years, the role of AGR2 in tumor development and progression has become more and more intensively studied [5,6]. The contribution of AGR2 to malignant transformation, drug resistance, and the development of metastasis has already been reported by various authors [7,8,9,10,11]. However, the mechanism of action, as well as the scope of AGR2 functions, remain unclear. Several reports described decreased AGR2 expression in tumor cells exposed to transforming growth factor beta (TGF-β) indicating its potential association with EMT, most probably via the SMAD4 signaling pathway [12,13,14]. We found previously that AGR2 levels positively correlate with the epithelial marker E-cadherin. Accordingly, reduction of AGR2, both physiologically by TGF-β and by gene knockout, was concomitant with the classical features of mesenchymal cells such as loss of E-cadherin, induction of N-cadherin, and morphological changes arising from cytoskeleton reorganization, including diffuse cytoplasmic distribution of vimentin and relocalization of F-actin [15]. Following these findings, we recently described an inverse correlation between AGR2 and ZEB1 (zinc finger enhancer binding protein, δEF1). We proposed the existence of a negative feedback regulatory mechanism through which ZEB1 binds to the *AGR2* promoter, thus repressing *AGR2* transcription. On the other hand, AGR2 negatively regulates ZEB1 levels, probably by controlling the stability of *ZEB1* mRNA [16].

AGR2 has an undeniable role in the tumorigenesis and metastasis formation of epithelial tumors. Therefore, to further determine how AGR2 influences signaling pathways in cancer, A549 derived lung cancer cell line clones with or without AGR2 expression and with or without TGF-β treatment were subjected to RNA sequencing (RNAseq). These data revealed important roles of AGR2 in the focal adhesion (FA) pathway and arachidonic acid metabolism. Independent RT-qPCR and immunochemical analysis confirmed these AGR2-dependent expression profiles. The potential role of AGR2 in arachidonic acid metabolism was additionally supported by the interdependent relation between AGR2 and prostaglandin E_2_ (PGE_2_).

## 2. Results

### 2.1. Transcriptome Analysis

Based on data showing the role of AGR2 in EMT [15,16,17,18], total RNA was isolated and sequenced to identify new determinants functionally associated with AGR2 during TGF-β induced EMT in A549 cells. Two biologically independent RNA sample sets consisting of A549 scr (scramble) cells and A549 KOAGR2 cells (cells with disrupted AGR2 expression), either exposed or not exposed to TGF-β were prepared and sequenced. Differentially expressed genes with adjusted *p*-value < 0.05 were included. According to this criterion, we acquired in total 1449 genes (Appendix A) that were subsequently analyzed according to their differential expression in relation to untreated A549 scr cells. To reflect that expression changes in some genes would be negligible and would devalue the functional annotation with signaling pathways, we set the fold-change (FC) threshold to identify genes that were altered by 1.5-fold or more (downregulated genes were defined as FC from 0.67 or lower, and upregulated genes as FC 1.5 or higher, compared to untreated A549 scr cells (Appendix A). We identified 361 genes with FC < 0.67 and 502 genes with FC > 1.5 in A549 scr cells exposed to TGF-β (Figure 1A). Importantly, AGR2 was identified among the downregulated genes, which independently supports our previous findings that TGF-β suppresses AGR2 expression [15]. We identified a larger number of genes in A549 KOAGR2 cells exposed to TGF-β (643 downregulated and 626 upregulated), which indicates an additive effect of combined AGR2 loss and TGF-β treatment. In untreated A549 KOAGR2 cells, 237 genes were downregulated and 138 were upregulated compared to untreated A549 scr cells, indicating that *AGR2* gene knockout had the lowest impact on gene expression changes but was associated predominantly with downregulation of gene expression in contrast to TGF-β treatment that predominantly associated with gene upregulation. 

We found that all three groups of samples (A549 scr TGF-β, A549 KOAGR2, and A549 KOAGR2 TGF-β) shared 96 downregulated genes and 61 upregulated genes compared to A549 scr cells grown in the absence of TGF-β (Figure 1A). The common genes were analyzed by Gene Ontology (GO) enrichment pathway using the Protein Analysis Through Evolutionary Relationships database (PANTHER visualization tools) [19,20]. The majority of both upregulated and downregulated genes were classified in cellular processes (Figure 1B). Therefore, we further analyzed the child terms of cellular processes. Downregulated genes were enriched for GO terms associated with metabolic pathways. Interestingly, upregulated genes were represented among GO terms such as cellular response to stimulus, cell communication, and signal transduction (Figure 1C). Statistical overrepresentation analysis showed that downregulated genes are involved in metabolic pathways. Although the spectrum of upregulated genes is higher, there is a clear enrichment of processes associated with EMT induction, such as regulation of cellular adhesion, migration, angiogenesis, and regulation of cellular signaling (Appendix A).

### 2.2. Functional Annotation 

To identify key cell signaling pathways associated with gene expression changes, we used DAVID Functional Annotation Clustering, a tool at the DAVID 2021 Knowledgebase [21,22]. DAVID bioinformatics resources consist of an integrated biological knowledgebase and analytic tools aimed at systematically extracting biological meaning from large gene/protein lists. As a proof of concept, we first analyzed all genes that displayed changes in their expression outside the FC threshold (FC < 0.67 or FC > 1.5) in response to TGF-β treatment in A549 scr cells endogenously expressing AGR2 (Figure 2A). We observed significant associations (*p* < 0.05) with many prominent cancer-associated signaling pathways, including focal adhesion, ECM-receptor interaction, PI3K-Akt signaling, Hippo signaling, small cell lung cancer, etc. (Appendix A). Interestingly, TGF-β signaling was the only pathway identified by both KEGG Pathway and BIOCARTA, indicating that this is the key pathway associated with expression changes observed in our model. 

Subsequently, we focused on signaling pathways influenced by AGR2 gene knockout (A549 KOAGR2). Our analysis included both upregulated and downregulated genes outside the FC threshold (Figure 2B). In total, 28 significantly associated pathways (*p* < 0.05) were identified (Appendix A), of which the focal adhesion pathway was the most significant. Altogether, 17 significant hits were clustered under this KEGG pathway. Interestingly, 10 genes showed increased expression after AGR2 gene knockout, as illustrated by the pathway scheme where these identified transcripts are depicted in green while downregulated genes are shown in red (Appendix A). These genes are predominantly involved in regulation of the actin cytoskeleton, implying extensive intracellular rearrangements in response to AGR2 loss (Table 1). Indeed, formation of actin stress fibers linked to vinculin-containing focal adhesions was observed in A549 cells after *AGR2* knockout, as opposed to A549 scr cells where vinculin is distributed diffusely in the cytoplasm (Figure 3A and additional slides in Appendix A). Vinculin foci were also observed in response to TGF-β treatment in both AGR2 positive and negative cells, with a more marked effect on AGR2 KO cells (Figure 3A and Appendix A). Stress fibers are contractile bundles governing migration and adhesion, and together with FA they form a mechanosensitive machinery [23]. This actin binding and bundling by vinculin is required for traction forces and hence is critical for cell migration [24]. Therefore, we also analyzed the invasive and migratory potential of these cells using transwell (Figure 3B) and wound healing assays (Figure 3C), respectively. As expected, scr untreated cells showed the lowest invasion and migration rate. Interestingly, *AGR2* knockout seems to play a more prominent role in migratory properties compared to TGF-β treatment in AGR2 expressing cells. ijms-23-10845-t001_Table 1Table 1The role and regulation of significant genes connected with focal adhesion in cells after *AGR2* knockout as clustered by DAVID analysis tool.GeneRegulationFunction***ROCK1***UPPromotes adhesion, migration, and invasion by inhibition of PTEN and activation of the PI3K/Akt signaling pathway through FAK phosphorylation. Overexpression is associated with invasive and metastatic phenotype [25].***COL4A1***UPShown to be overexpressed by a number of malignancies, including hepatocellular, urothelial, or pancreatic cancer, where it contributes to migratory and invasive phenotype [26,27,28].***COL4A2***UPUpregulated in pre-neoplastic and HCC tissue and overexpression correlated with shorter progression survival [29]. Reported to promote cell adhesion, migration, and proliferation of different cell types [30,31].***FLNA***UPActin crosslinking protein with an important function in migration and adhesion [32]. Showed to be suppressed by miR-200c independently of ZEB1 [33].***MYLK***UPAlso known as MLCK, contributes to adhesion, migration, invasion, and metastasis [34,35] and represents a miR-200c target [36]. Phosphorylates myosin light chain, which facilitates the association of myosin with F-actin and hence generates contractile forces in HCC [37].***PAK3***UPSMAD4 effector mediating metastatic signals through the PAK3-JNK-Jun pathway in NSCLC [38].***TNC***UPAn ECM glycoprotein inducing EMT and activating Src and FAK [39,40]. In Pancreatic cancer, it activates JNK/C-Jun pathway leading to production of MMP-9 and induced paxillin phosphorylation and FA formation [41].***VEGFA***UPWell-known mediator of angiogenesis and a regulator of proliferation, survival, adhesion, migration, and invasion [42].***VCL***UPRegulates polarized migration, controls integrin activity through interaction with talin, regulates recruitment and diffusion of core FA proteins in a force-dependent manner, and is required for an efficient cellular adhesion and migration [43].***ZYX***UPOne of the key FA proteins contributing to migration, invasion, adhesion, and proliferation [44,45]. Induced in A549 by TGF-β through Smad3 [46].***COL4A4***DOWNComponent of basement membrane and a component of focal adhesion (according to WikiPathways and KEGG pathways).***FN1***DOWNSeemingly conflicting role in cancer and metastasis, which is well discussed in a review by Lin and colleagues [47].***ITGB4***DOWNTranscriptionally repressed by ZEB1 [48].***ITGB5***DOWNPlays a role in TGF-β induced EMT [49] and facilitation of migration in HCC [50].***ITGB8***DOWNExclusive heterodimerization with αv subunit—αvβ8 serves as a receptor for latent TGF-β and activates it in protease dependent manner [51]. αvβ8-TGF-β axis mediates cell-cell communication and its dysregulation leads to aberrant adhesion and signaling [52]. Interacts with FAK, which activates VAV and RAC1 in endometrial epithelial cells [53].***LAMA3***DOWNPart of laminins, which are further composed of α and γ subunits. *LAMA3* encodes 2 different transcripts, LAMA3A and 3B. Laminin 332 containing laminin α3A was described as a regulator of cell migration, and in focal adhesion α3A interacts with integrin α3β1, which associates with signaling molecules and connects to the actin cytoskeleton through linker molecules. Further details are available in the review by Hamill et al. [54].***VAV3***DOWNActs as an EMT promoter together with ZEB1 [55].

Several lines of evidence indicate that TGF-β attenuates AGR2 expression in human cancer cells [12,14,15,16]. Thus, identifying the expression patterns common for both AGR2 gene knockout and the effect of TGF-β may reveal crucial signaling pathways and/or cellular processes driven by TGF-β via regulation of AGR2. We selected genes showing significant expression changes outside the FC threshold (FC < 0.67 or FC > 1.5) that were common for both A549 KOAGR2 cells and A549 scr cells exposed to TGF-β (Figure 2C). In total, 158 genes (62 upregulated and 96 downregulated) were analyzed by DAVID functional annotation tool, which similarly to the case of the analysis of KOAGR2 cells (Appendix A), identified arachidonic acid metabolism pathway, with the same down-regulated genes (Table 2, Appendix A). Interestingly, when we separately analyzed downregulated or upregulated genes, the *p*-value of arachidonic metabolism determined for downregulated genes became more significant (*p* < 0.01), and three additional genes were assigned to the metabolism of eicosanoids (*p* < 0.01), which represent arachidonic acid-derived lipid mediators (Appendix A) [57]. In contrast, upregulated genes were associated with focal adhesion (six genes, *p* < 0.01), PI3K-AKT signaling (seven genes, *p* < 0.01), and “proteoglycans in cancer” (six genes, *p* < 0.01). Taken together, these data are in agreement with transcriptomic profiling of cells with abrogated AGR2 expression (A549 KOAGR2) and support the synergistic effect of *AGR2* knockout and TGF-β treatment. 

In line with previous findings, exposure of A549 KOAGR2 cells to TGF-β (Figure 2D) resulted in the identification of 60 signaling pathways, with focal adhesion being the most significant (Appendix A). The relatively large number of genes (1270) showing significant changes corresponds with the quantity of signaling pathways potentially influenced by TGF-β exposure of cells without AGR2 expression. 

Similar to GO annotations in PANTHER (Figure 1), we also applied analysis in DAVID, showing the same trend in their expression patterns (i.e., up- or down-regulation) after TGF-β treatment, *AGR2* knockout, as well as their combination in relation to untreated A549 scr cells (61 upregulated, 96 downregulated genes, threshold FC < 0.67 or FC > 1.5, Figure 2E). In total, 19 signaling pathways were identified, with pathways in cancer and proteoglycans in cancer as the two most prominent hits, indicating that AGR2 downregulation and TGF-β exposure, either alone or combined, provide protumoric and prometastatic signals (Appendix A). Furthermore, FA pathway and arachidonic acid metabolism were other significant KEGG pathways that are in accordance with PANTHER results where metabolic processes were shown to be downregulated, and statistical overrepresentation also showed prostaglandin biosynthesis to be downregulated (Appendix A). In addition, adhesion, wound healing, chemotaxis, and cell migration were all upregulated in PANTHER (Appendix A). Lastly, MAPK signaling is another prominent hit in DAVID analysis, in accordance with upregulated MAPK cascade from PANTHER.

### 2.3. RNAseq Data Validation

To support our RNAseq data, expression of selected genes was independently validated using RT-qPCR from newly prepared samples. Six representative genes linked with focal adhesion (*COL4A1*, *COL4A2*, *FLNA*, *VAV3*, *VEGFA*, and *VINC*) (Figure 4A) and three involved in arachidonic acid metabolism (*GPX2*, *PTGS2*, *PLA2G4A*) were determined (Figure 4B). The RT-qPCR results reflect the RNAseq data in both pathways. The changes in mRNA levels of focal adhesion associated genes were all significant after TGF-β treatment. In *AGR2* knockout cells, the mRNA levels behaved similarly to RNAseq data, where the changes in mRNA levels are also lower compared to the effect of TGF-β. However, these changes were mostly non-significant using RT-qPCR indicating that *AGR2* knockout alone is not crucial for FA regulation, but rather shows a synergistic effect in combination with TGF-β as shown for *COL4A1*, *COL4A2*, *VAV3*, and *VINC* (Figure 4A). Interestingly, *AGR2* gene knockout as well as exposure to TGF-β led to a significant decrease in the mRNA levels of all three genes associated with arachidonic acid metabolism (Figure 4B). Taken together, quantitative PCR confirmed changes in the expression patterns of selected genes involved in these signaling pathways. We also analyzed several transcripts at the protein level (Figure 4C). COL4A1 and PTGS2 (COX-2) show homologous protein levels to their respective mRNA levels. No change was seen for VAV3.

Additionally, we included colorectal cancer cells HT-29 as another cell line for validation. RT-qPCR results reflect the RNAseq data measured in A549 for both pathways (Appendix A). Concerning the protein level changes, VAV3 was not changed between samples, COL4A1 is induced after TGF-β treatment in scr samples and is lost in KOAGR2, and PTGS2 (COX-2) is downregulated similar to A549 cells (Appendix A).

### 2.4. The Role of AGR2 in Relation to Arachidonic Acid Metabolism and Prostaglandin E2 Biosynthesis

Prostaglandin E_2_ (PGE_2_) is a bioactive lipid from the prostanoid family and one of the primary products of arachidonic acid metabolism that is connected with many physiological and pathological conditions. Following our data identifying arachidonic acid metabolism as the most prominent pathway associated with AGR2 gene knockout as well as TGF-β dependent attenuation of AGR2 expression, we determined PGE_2_ levels in cell culture media by ELISA. There was a slight decrease in PGE_2_ level in the culture media of A549 scr cells exposed to TGF-β and a substantial decrease in the culture media in which A549 AGR2KO cells were maintained (Figure 5A). These results functionally confirm our transcriptomic data and indicate a direct involvement of AGR2 in PGE_2_ biosynthesis. 

Cyclooxygenases (COX, prostaglandin endoperoxide synthases) play an irreplaceable role in the conversion of arachidonic acid to prostaglandins. In parallel, these enzymes are promising therapeutic targets, especially for colorectal and lung cancer [62,63,64,65]. Therefore, in addition to the lung adenocarcinoma cell line A549, we also analyzed the potential relationship between AGR2 and cyclooxygenases in the colorectal carcinoma cell line HT-29. These enzymes are represented by two isoforms, COX-1 and COX-2, catalyzing the conversion of arachidonate to prostaglandin H_2_, the substrate for a series of cell-specific prostaglandin and thromboxane synthases, including prostaglandin E synthase responsible for PGE_2_ generation [66]. To elucidate the relationship between COX and AGR2, A549 and HT-29 cells were exposed to PGE_2_ and COX inhibitors. AGR2 protein was induced in both cell lines exposed to PGE_2_ (Figure 5B). Nevertheless, no significant decrease in AGR2 was detected after treatment with COX inhibitors. 

To support the involvement of AGR2 in arachidonic acid metabolism, we screened publicly accessible colorectal and lung cancer gene expression datasets (Cancer Cell Line Encyclopedia and COSMIC) and correlated *AGR2* mRNA levels with expression of genes directly involved in prostaglandin biosynthesis. The results were subsequently visualized via heat maps (Appendix A) and interdependence was examined via Pearson correlation coefficients (Table 3). These results show interdependence between AGR2 and PLA2G2A, PLA2G4A with positive correlation. The product of the *PLA2G2A* gene promotes prostaglandin E_2_ synthesis that stimulates Wnt signaling and has been identified as a susceptibility gene for cancers of the small and large intestine [67]. *PLA2G4A* encodes a member of the cytosolic phospholipase A2 group IV family enzyme catalyzing the hydrolysis of membrane phospholipids to release arachidonic acid, which is subsequently metabolized into eicosanoids [68]. It is an important enzyme in tumor development including colorectal and lung cancer [69,70]. In contrast, genes encoding members of the cytochrome P450 superfamily of enzymes CYP2B6, CYP4F2, and CYP4F3 showed negative correlations with AGR2 expression, as did *PLA2G12B*, expression of which is frequently downregulated in tumors, suggesting that *PLA2G12B* is a negative regulator of tumor progression [71]. 

## 3. Discussion

The improvement of molecular biology and omics technologies in the past two decades has significantly impacted the course of cancer research and sped up progress in biomarker discovery. In particular, the substantial advances in high-throughput technologies and multi-omics strategies have enabled the identification of new cancer biomarkers that can predict patients’ responses to treatment and prognoses, leading to the development and/or improvement of personalized medicine. This is raising the hope of increased efficacy in the diagnosis and treatment of malignant diseases. Nevertheless, a huge influx of untargeted omics datasets clearly showed that the identification and validation of potential biomarkers is a task that needs to be robust and reproducible and is commonly comprised of biological and computational processes driven by multi-disciplinary collaborations between many groups, including basic cancer researchers, technologists, clinical oncologists, drug companies, and other healthcare professionals.

In our preclinical research, we performed detailed analysis of the role of AGR2, currently showing as an important tumor marker [5,72]. Its elevated expression occurs in a number of malignancies compared to healthy tissue and, for instance, AGR2 is a marker of poor prognosis in breast and prostate cancer [11,73,74,75]. The role of AGR2 has been described in many cellular processes, including proliferation, DNA stability, cell death, drug resistance, ER stress, angiogenesis, and adhesion [10]. The precise role of AGR2 in a number of these processes still evades full understanding, although substantial evidence points towards its important role in cancer progression and metastasis.

Based on our previous research uncovering AGR2 as a keeper of epithelial phenotype [15,16], we aimed to analyze how *AGR2* knockout and/or TGF-β treatment influence gene expression by RNA sequencing. Following the loss of AGR2 expression with or without TGF-β pathway as one of the most prominent alterations. *AGR2* knockout on its own led to significant changes in genes that cluster into focal adhesion, ECM-receptor interaction, and regulation of actin cytoskeleton pathways, which all converge on the hypothesis that knockout of AGR2 influences cell migration and invasion.

Focal adhesions are a type of ECM-cell adhesion structure that connects integrins to the actin cytoskeleton. However, these are not only static connection points between the cell cytoskeleton and ECM, rather they are more of a communication canal through which ECM influences cellular processes and integrins act as bi-directional signal passageways in this communication. Most of the signals converge to FAK and Src kinases, which regulate a plethora of cellular pathways including PI3K/AKT and MAPK/ERK representing some of the most prominent pathways in cancer progression [76,77,78]. FAs are multi-protein complexes including adaptor/scaffolding proteins such as vinculin, which is an F-actin binding protein. Vinculin is known to regulate polarized migration, control integrin activity through interaction with talin, regulate recruitment and diffusion of core FA proteins in a force-dependent manner, and is required for efficient cellular adhesion and migration [43]. Our analysis identified VINC as one of the upregulated transcripts in cells after *AGR2* knockout exposed to TGF-β treatment. More importantly, we also show vinculin localization into foci after either KOAGR2 or TGF-β treatment, implying the connection of AGR2 to the FA pathway and that the absence of AGR2 could lead to a similar cellular response as TGF-β signaling. However, changes in the level of transcripts used for FA pathway validation were mostly non-significant in *AGR2* knockout cells and were significant only after treatment with TGF-β alone or in combination with KOAGR2. Therefore, the interpretation of our results requires caution, and it seems that AGR2 downregulation may act more strongly as a downstream effector of TGF-β in the focal adhesion pathway leading to an enhanced cascade of signaling events favoring cancer progression and dissemination. On the other hand, KOAGR2 cells showed higher invasive potential using transwell assay and higher migration rate than A549 scr cells. These results support our previous work showing in A549 cell line that KOAGR2 indeed has a pro-migratory and EMT-promoting effect [15]. Moreover, independent database mining [79] also showed that *AGR2* mRNA is positively correlated with the expression of epithelial genes and inversely correlated with mesenchymal genes in carcinoma cell lines of various origins, supporting its role as an epithelial marker. However, the precise involvement of AGR2 in the pro-migratory pathways and its role in the FA formation and turnover warrants deeper analysis.

In contrast to the upregulation of components of actin reorganization and the FA pathway, *AGR2* knockout reduced the expression of genes associated with arachidonic acid metabolism, indicating the involvement of AGR2 in this biosynthetic pathway and eicosanoid metabolism. Arachidonic acid is stored as glycerophospholipids, which compose the lipid bilayer of the plasma membrane, and its release is usually mediated by phospholipases of the A2 type, optionally phospholipases C and D, resulting in numerous mediators with roles in a wide range of physiological and pathological processes [80]. The positive correlation of AGR2 with PLA2G4A, encoding cytosolic phospholipase A_2_, was identified not only by our analysis but also by data mining in independent databases. Arachidonic acid released by cytosolic phospholipase A_2_ is metabolized by COX-1 and COX-2 to generate eicosanoids, including prostaglandins, thromboxanes, and prostacyclins. The association of AGR2 with arachidonic acid metabolism and the positive impact of AGR2 on PGE_2_ production (Figure 5) encouraged us to study the potential relationship between COX-2 and AGR2. Both genes are frequently overexpressed in epithelial malignancies, including colorectal and lung cancer, and are associated with poor clinical outcomes. In addition, NSAIDs targeting COX-2 or PGE_2_ have been reported to have a protective effect against the development of colorectal cancer [64,81,82,83]. However, no significant effect of nonsteroid anti-inflammatory drugs on AGR2 expression was observed in tested cell lines. Nevertheless, Zhang et al. recently showed that *AGR2* knockdown enhances the therapeutic effects of a COX-2 inhibitor, celecoxib, in CRC metastasis [84]. Interestingly, we observed that addition of exogenous PGE_2_ elevated AGR2 protein in A549 lung carcinoma cells, with a similar trend observed in the colorectal cancer cell line HT-29. These findings are also supported by experiments identifying the EP4-PI3K-AKT pathway as a mediator of PGE_2_ dependent induction of AGR2 and its indispensable role in PGE_2_-induced CRC metastasis [84]. However, the EMT-promoting role of AGR2 described by Zhange et al. is contradictory to our results describing AGR2 as a keeper of the epithelial phenotype [15]. In our present analysis, we show that KOAGR2 induces downregulation of arachidonic acid metabolism resulting in PGE_2_ downregulation in media while the addition of PGE_2_ induces AGR2 levels. There is an undeniable role of PGE_2_ in a plethora of cellular processes including tumor formation, proliferation, angiogenesis, inflammation, immune surveillance, apoptosis evasion, adhesion, and migration [85,86]. EP receptors initiate downstream signaling through coupled G proteins leading to Ca^2+^, cAMP, IP3, which ultimately leads to transcription induction and crosstalk with other cellular pathways [87,88]. Therefore, PGE_2_ exerts a complex and multipurpose role in malignancies and AGR2 may mediate some of its functions, for instance proliferation [89,90]. Moreover, the subcellular localization of AGR2 is crucial and thus the ability of some cells to secrete AGR2 opens up possibilities for a wider dual signaling role of this protein [91,92]. In parallel, the role of PGE_2_ in regulating the inflammatory milieu that drives cancer onset and progression could also be connected with induced AGR2 expression in malignant cells. A recent publication has shed more light on the potential mechanism responsible for AGR2 induction during inflammation by demonstrating that AGR2 dimers act as sensors of ER homeostasis and can be disrupted upon ER stress (caused by imbalance in AGR2 client proteins). This leads to the secretion of AGR2 monomers that act as systemic alarm signals for pro-inflammatory responses, resulting in uncontrolled inflammation [93].

In conclusion, our datasets provide a comprehensive insight into the transcriptome of A549 lung cancer cells after manipulation of AGR2 expression and/or TGF-β treatment, which extends our previous findings describing the role of AGR2 in the lung cancer model. Furthermore, our data support previously published results that AGR2 indeed has a prominent role in cellular adhesion, providing more detailed information about its related pathways. At the same time, we have identified AGR2 as an important component of the arachidonic acid metabolic pathway, filling a gap in understanding how AGR2 is involved in inflammatory processes.

## 4. Materials and Methods

### 4.1. Cell Lines and Reagents

Lung cancer cell line A549 (ATCC^®^ CCL-185™) and colorectal cancer cell line HT-29 (ATCC^®^ HTB-38™), originating from American Type Culture Collection (ATCC, Manassas, VA, USA) were maintained in high glucose Dulbecco’s Modified Eagle’s Medium (DMEM, Sigma-Aldrich, St. Louis, MO, USA) supplemented with 10% FBS (Life Technologies, Carlsbad, CA, USA), 1% pyruvate, and L-glutamine at 37 °C in a humidified atmosphere of 5% CO_2_. Unless otherwise stated, both cell lines were grown to 70–80% confluence prior to treatment. TGF-β1 (R&D Systems, Minneapolis, MN, USA) was added to a final concentration of 1 ng/mL for 24 h for RNA sequencing library samples and 5 ng/mL for the RT-qPCR and IFC. PGE_2_ was used in a final concentration of 1 μM, Diclofenac 10 μM, Celecoxib 10 μM, acetylsalicylic acid 10 μM (all Merck, Darmstadt, Germany).

Cell lines with *AGR2* gene knockout were prepared as described previously using CRISPR/Cas9 [15]. Briefly, A549 and HT-29 cells were transfected with plasmid LentiCRISPR-v2_AGR2 or LentiCRISPR-v2_scrambled serving as a control and cells were exposed to puromycin for several weeks. Clones were selected from the pool of resistant cells and tested for AGR2 expression and validated by sequencing. Two clones were used both for A549 and HT-29 cells: A549 scr and HT-29 scr (scrambled, i.e., with AGR2 expression) and A549 KOAGR2 and HT-29 KOAGR2 (without AGR2 expression).

### 4.2. RNA Purification and Sequencing

Total RNA was extracted from A549 cells by TRI-Reagent (MRC). Only RNA samples with RNA integrity number (RIN) ≥ 7 determined by Bioanalyzer (RNA 6000 Nano Kit, Agilent, Santa Clara, CA, USA) passed to library preparation. The TruSeq Stranded Total RNA LT Sample Prep Kit (Illumina, San Diego, CA, USA) was used to convert 0.5 μg of total RNA into a library of template molecules. The library was validated using Bioanalyzer (DNA 1000 Kit, Agilent, Santa Clara, CA, USA) and quantified according to the manufacturer’s instructions by qPCR (KAPA Library Quantification Kit Illumina platforms, Kapa Biosystems, Wilmington, MA, USA) using Quant studio (QuantStudio 5, Thermo Fisher Scientific, Waltham, MA, USA). Samples were sequenced using NextSeq 500 (Illumina, San Diego, CA, USA). Low-quality reads were removed from the raw sequencing data and adaptor sequences clipped as well as leading or trailing regions of low quality (below 18 phred) using Trimmomatic 0.33 and bbduk2 packages. Reads were mapped to the reference genome using STAR 2.5.3; the reference sequence version used was b37 from GATK (derived from GRCh37) with GENCODE release 24 annotation. Gene coverage was calculated using HTSeq-count. Differential expression was evaluated using DESeq2.

### 4.3. Data Analysis, Statistical Analysis and Data Mining

To facilitate biological interpretation in a cellular signaling network context, we used data from KEGG and BioCarta pathways that are DAVID 2021 Knowledgebase defined defaults. In the Functional Annotation Chart, just BBID, BIOCARTA, and KEGG_PATHWAY were selected. Default settings were tightened to count threshold 3 and EASE score (a modified Fisher exact *p*-Value) < 0.05. Genes for analysis of interdependence between AGR2 and metabolism of arachidonic acid were selected based on the pathway of Arachidonic acid metabolism from the KEGG database. Data for gene expression in colorectal and lung cell lines and tumours were mined from Cancer Cell Line Encyclopedia and COSMIC, respectively [94,95]. Results were visualized via heat map and interdependence was examined via Pearson correlation coefficient.

### 4.4. Gene Expression

M-MLV Reverse Transcriptase (Sigma-Aldrich, St. Louis, MO, USA) was used to reverse transcribe total RNA extracted from cells using TRIzol reagent (Sigma-Aldrich, St. Louis, MO, USA). Either SYBR Green MasterMix (Roche, Basel, Switzerland) or TaqMan Universal PCR MasterMix (Life Technologies, Carlsbad, CA, USA) were used for quantitative PCR. *HPRT1*, 18*S* rRNA and *GAPDH* served as parallel endogenous controls. The data represent means of three technical triplicates within each independent biological replicate (n = 3). The primer sequences are listed in Appendix A. The relative mRNA expression levels of each gene were calculated using the 2^−ΔΔCT^ method. The statistical significance was calculated using the ordinary One-Way ANOVA test with Post Hoc Tukey HSD.

### 4.5. Western Blot Analysis

Cells were washed twice with cold phosphate-buffered saline (PBS) and then scraped into NET lysis buffer (150 mM NaCl, 1% NP-40, 50 mM Tris pH 8.0, 50 mM NaF, 5 mM EDTA pH 8.0) supplemented with protease and phosphatase inhibitor cocktails according to the manufacturer’s instructions (Sigma-Aldrich-St. Louis, MO, USA). Following SDS-PAGE, samples were transferred onto nitrocellulose membranes and incubated overnight at 4 °C with primary antibodies. The following day, membranes were washed and probed with horseradish peroxidase (HRP)-conjugated secondary antibodies (1:1000) for 1 h at room temperature (RT). Chemiluminescent signals were developed using ECL solution and visualized with GeneTools (Syngene). Actin was used as a loading control and as a reference for normalization. Antibodies: AGR2 rabbit polyclonal serum (K-31, in-house); actin monoclonal antibody (ACTN05 C4), COL4A1 Polyclonal Antibody (PA585634), VAV3 Polyclonal Antibody (PA5113523) (all ThermoFisher Scientific, Waltham, MA, USA), PTGS2 antibody (HPA001335, Sigma-Aldrich, St. Louis, MO, USA), HRP-conjugated swine anti-rabbit, and HRP-conjugated rabbit anti-mouse (both Dako, Agilent, Santa Clara, CA, USA). Statistical significance was calculated using One-Way ANOVA test with Post Hoc Tukey HSD.

### 4.6. Immunofluorescent Staining

Cells were seeded onto sterile coverslips and treated with 5 ng/mL TGF-β for 24 h. Cells were then washed with PBS and fixed with 4% formaldehyde (Sigma-Aldrich, St. Louis, MO, USA) in PBS for 20 min at RT. After washing with PBS, cells were permeabilized with 0.2% Triton X-100 in PBS for 5 min at RT. Permeabilised cells were washed again with PBS and blocked for 30 min with 3% BSA (Sigma-Aldrich, St. Louis, MO, USA) in PBS-Tween. Staining with anti-Vinculin antibody FAK kit (Sigma-Aldrich, St. Louis, MO, USA) was done at dilution 1:100 for 1 h at 37 °C. Stained cells were washed three times with PBS and incubated with secondary Alexa Fluor 488 goat anti-mouse IgG (Abcam, Cambridge, UK), F-actin Phalloidin (Abcam, Cambridge, UK), and Hoechst (Sigma-Aldrich, St. Louis, MO, USA) for nuclei staining for 1 h at RT. Coverslips were washed three times with PBS, once with miliQ water and mounted using VECTASHIELD mounting medium (Vector Laboratories, Newark, CA, USA). Samples were photographed on an Olympus BX41 microscope (Olympus) at 100× magnification using immersion oil (ibidi). Vinculin foci were quantified according to Horzum et al. using ImageJ software [56,96]. Briefly, raw fluorescent images were processed by the SLIDING PARABOLOID option with the ROLLING BALL radius set to 50 pixels followed by CLAHE plugin (blocksize = 19, histogram = 256, maximum = 6), mathematical EXP, manually setting BRIGHTNESS and CONTRAST to automatic, running the Mexican Hat filter plugin [97] with radius = 5, running the TRESHOLD command with default method and automatic adjustment and lastly executing the ANALYZE PARTICLES command (size = 50, Infinity circularity = 0.00–0.99 show = Outlines display clear summarize). At least 12 individual slides were analyzed with a total of at least 79 individual cells per sample. The analysis provides total count of foci per image. We divided the count by the number of cells in each image to obtain “count per cell” of each picture. These values were then averaged for all images from each condition to get the “average count per cell” and their corresponding standard deviation. The analysis also provides values for the total area of foci per image in pixels squared. We divided that number by the number of cells in each picture to obtain the “total area per cell” and likewise to foci the “average total area per cell” in pixels squared. Statistical significance was calculated using One-Way ANOVA with Post Hoc Tukey HSD.

### 4.7. Invasion and Migration Analysis

The invasion was analyzed using the CytoSelect™ 24-Well Cell Migration and Invasion Assay (8 µm, Colorimetric Format) from Cell Biolabs (San Diego, CA, USA) according to the manufacturer’s instructions. Briefly, 500,000 cells were added to the upper chamber pre-coated with basement membrane in triplicates in serum free media and left for 24 h in cell culture incubator. Medium with 10% FBS was used as chemoattractant. Afterwards, the non-invading cells were removed with cotton swabs and the cells that invaded to the bottom of the membrane were fixed and stained with staining solution. The stained cells were dried and dissolved with extraction solution. Results are represented as OD 560 nm using spectrophotometry (Tecan, Männedorf, Switzerland).

Migration was analyzed using the IncuCyte^®^ Scratch Wound 96-Well Real-Time Cell Migration assay (Sartorius, Göttingen, Germany). A total of 50,000 cells were seeded per well in quadruplicates per condition. The following morning, cells were scraped using the wound making tool, washed, and incubated in serum free medium with or without TGF-β for 24 h inside the IncuCyte machine, which scanned the plate every 4 h. The results were analyzed by the integrated metric “Relative wound density”. This metric measures spatial cell density in the wound area relative the outside of the wound area at every time point and as such is self-normalizing if any changes were to occur outside the wound, for instance due to changes in the proliferation or survival of cells.

### 4.8. PGE_2_ ELISA

In total, 2 × 10^5^ A549 scr and KOAGR2 cells were plated and incubated for 24 h and the medium was collected and centrifuged at 8000 rpm/RT. PGE_2_ concentrations were determined by ELISA (514010, Cayman Chemical, Ann Arbor, MI, USA) according to the manufacturer’s protocol. Statistical significance was calculated using One-Way ANOVA with Post Hoc Tukey HSD.

## Figures and Tables

**Figure 1 ijms-23-10845-f001:**
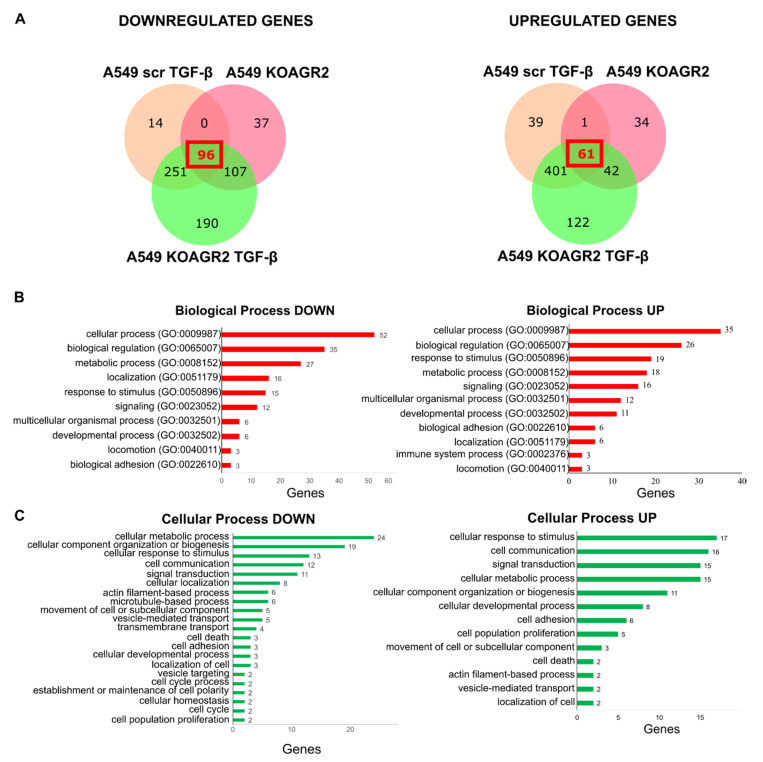
Evaluation of gene expression changes with respect to particular biological processes using the Protein Analysis Through Evolutionary Relationships database (PANTHER visualization tools). (**A**) Venn diagrams for overlapping genes with significantly decreased (**left**) or increased (**right**) expression in comparison with A549 scr cells. (**B**,**C**) GO enrichment analysis of biological processes (red graphs) and cellular processes (green graphs) associated with downregulated (**left**) and upregulated (**right**) genes common to all three categories.

**Figure 2 ijms-23-10845-f002:**
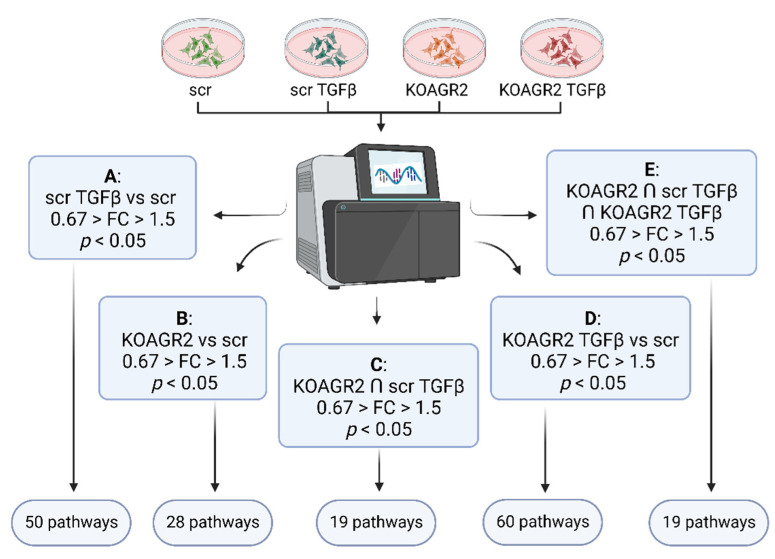
Schematic representation of samples subjected to RNAseq and the data analysis workflow using DAVID Functional Annotation Clustering tool at the DAVID 2021 Knowledgebase. (**A**) DAVID clustering of transcripts with significant FC in scr TGF-β samples vs scr control samples. (**B**) DAVID clustering of significant transcripts from KOAGR2 samples vs scr samples. (**C**) DAVID clustering of significant transcripts common between KOAGR2 and scr TGF-β. (**D**) DAVID clustering of significant transcripts from KOARG2 TGF-β vs scr. (**E**) DAVID clustering of significant transcripts common for KOAGR2, scr TGF-β, and KOAGR2 TGF-β. FC, fold change.

**Figure 3 ijms-23-10845-f003:**
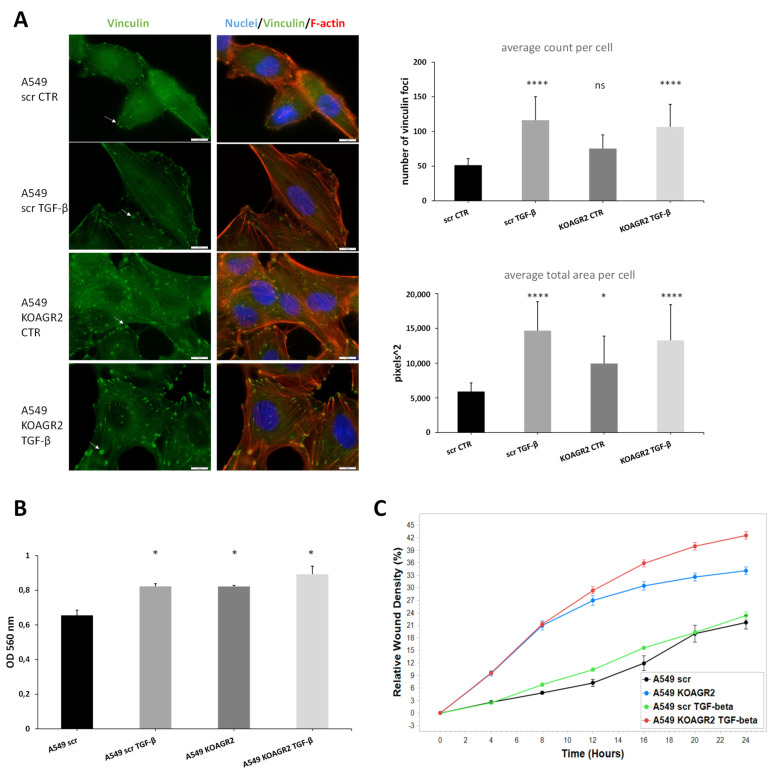
(**A**) Representative immunofluorescence of F-actin stress fibres (red) and vinculin foci (green). Nuclei are stained in blue by Hoechst. Slides were captured at 100× magnification and the scale bar represents 10 μm. Graphs show quantification of Vinculin foci either as average count per cell or average area taken up by foci per cell in pixels squared. Quantification was performed according to Horzum et al. [56]. (**B**) Analysis of invasion potential of A549 using transwell assay. (**C**) Analysis of migratory properties by wound healing assay using IncuCyte. The migratory rate is expressed as the relative wound density in % and is the highest in cells with *AGR2* knockout. * *p* < 0.05; **** *p* < 0.0001; ns (non-significant).

**Figure 4 ijms-23-10845-f004:**
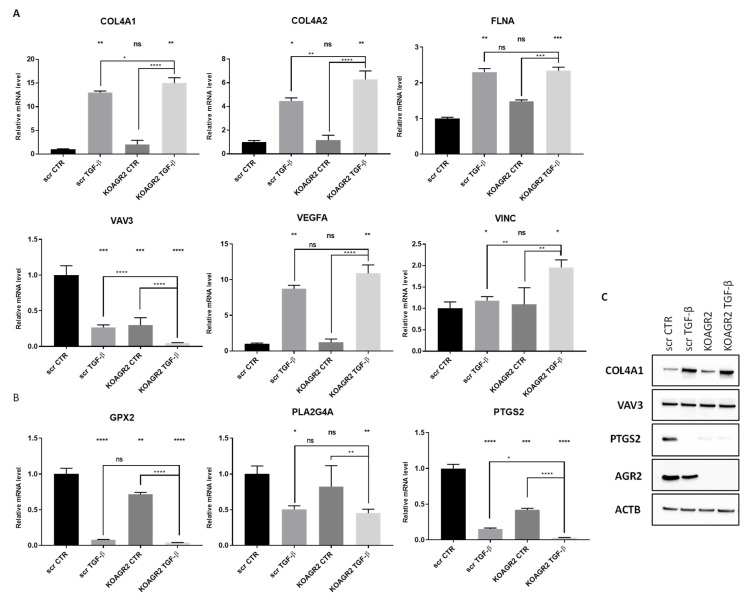
Validation of mRNA levels of representative genes by RT-qPCR for (**A**) focal adhesion pathway and (**B**) arachidonic acid metabolism pathway. Graphs show gene expression normalized to GAPDH as an endogenous control. In parallel, another two endogenous controls, 18*S* rRNA and HPRT1, were used with similar outputs. * *p* < 0.05; ** *p* < 0.01; *** *p* < 0.001; **** *p* < 0.0001; ns (non-significant). (**C**) Representative immunochemical analysis of COL4A1, VAV3, PTGS2, and AGR2. Beta-actin served as loading control.

**Figure 5 ijms-23-10845-f005:**
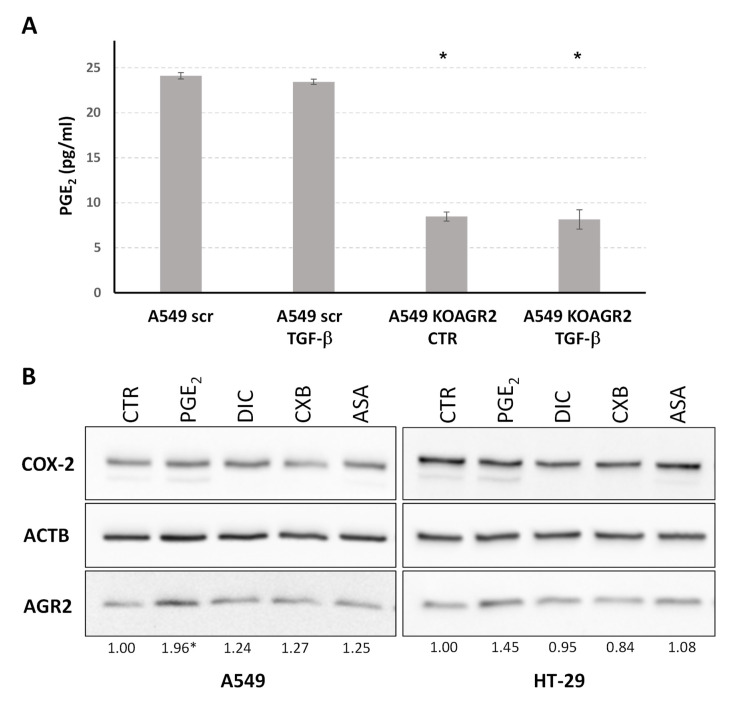
(**A**) ELISA determination of PGE_2_ released from A549 cells into culture media. (**B**) Representative immunochemical analysis of COX-2 and AGR2. Beta-actin served as a loading control and for normalization to determine fold changes calculated by densitometric analysis in cells exposed to prostaglandin E_2_ (PGE_2_), diclofenac (DIC), celecoxib (CXB), and acetylsalicylic acid (ASA) compared to untreated cells (CTR). Significant changes (*p* < 0.05) determined from three independent biological experiments are indicated by asterisk.

**Table 2 ijms-23-10845-t002:** The role and regulation of genes connected with arachidonic acid metabolism in cells after *AGR2* knockout and in response to TGF-β treatment as clustered by DAVID analysis tool.

Gene	Regulation	Function
*PLA2G4A*	DOWN	Hydrolysis of membrane phospholipids, which leads to release of arachidonic acid that is further metabolized into eicosanoids through one of three pathways (COX, LOX, CYP450) [58].
*PTGS2*	DOWN	Also known as COX-2, a rate-limiting enzyme involved in the conversion of arachidonic acid into prostaglandins [58].
*PTGES*	DOWN	Enzyme catalyzing conversion of COX derived PDH_2_ into PGE_2_ [59].
*AKR1C3*	DOWN	Downstream of COX catalyzes reduction of PGH_2_ and PGD_2_ into PGF_2__α_ [60].
*GPX2*	DOWN	Reduces fatty acid-derived hydroperoxides and inhibits NF-κB activity [61].

**Table 3 ijms-23-10845-t003:** Pearson correlation coefficients of interdependence between AGR2 and genes of arachidonic acid metabolism. Genes with Pearson coefficient of more than 0.5 (positive correlation) or less than −0.5 (negative correlation) are highlighted in bold.

Gene	Lung Tumors	Large Intestine Tumors	Lung Cell Lines	Colorectal Cell Lines
*ALOX12BA*	−0.154	0.330	−	−0.214
*ALOX15*	0.053	−0.324	−	−
*CBR3*	0.052	0.391	0.113	−0.137
** *CYP2B6* **	−0.053	**−0.622**	−	−
*CYP2C18*	0.341	0.427	−	−
*CYP2C9*	0.355	0.245	−	−
** *CYP4F2* **	0.060	**−0.576**	−	−
** *CYP4F3* **	0.363	**−0.599**	−0.046	−0.088
*DHRS4*	0.036	0.471	−0.097	0.165
*EPHX2*	0.305	−0.082	0.317	0.066
*GGT1*	0.040	−0.445	0.021	0.072
*LTC4S*	0.053	−0.395	−	−
*PLA2G10*	0.277	0.156	0.315	0.150
*PLA2G12A*	0.103	0.253	0.492	0.334
** *PLA2G12B* **	−0.022	**−0.622**	−	−
*PLA2G2A*	0.094	0.462	−	−
*PLA2G3*	0.093	0.347	−	−
** *PLA2G4A* **	0.068	**0.569**	0.176	0.264
*PLA2G6*	0.226	−0.091	0.255	0.474
*PTGS2*	−0.190	0.332	0.017	−0.074

## Data Availability

KEGG database https://www.genome.jp/kegg/, COSMIC database https://cancer.sanger.ac.uk/cosmic/downloaddata, Cancer Cell Line Encyclopedia https://sites.broadinstitute.org/ccle/ and data link: http://ftp.ebi.ac.uk/pub/databases/microarray/data/atlas/experiments/E-MTAB-2770/.

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
