# Peer review of "Identification of AGR2 Gene-Specific Expression Patterns Associated with Epithelial-Mesenchymal Transition"

_ijms, 2022, doi:10.3390/ijms231810845_

Round 1
Reviewer 1 Report
Manuscript ID: ijms-1839100
Type of manuscript: article
Title: Identification of AGR2 gene-specific expression patterns associated with epithelial-mesenchymal transition
Journal: International Journal of Molecular Sciences
In this article, the authors investigated the AGR2, an important protein of the cellular machinery responsible for maintaining the epithelial phenotype, which interferes with TGF-β pathway with manifested particularly in cancer cells by the induction of mesenchymal phenotype.
In particular, they performed transcriptomic profiling of a lung cancer cell line with CRISPR-Cas9 mediated AGR2 knockout together with and without TGF-β treatment. The obtained data were subjected to Bioinformatics Microarray Analysis. It was identified significant changes in transcripts linked with focal adhesion and eicosanoid production, in particular arachidonic acid metabolism.
The paper is well organized and well written and represents a contribution to the field of the pathway regulated by TGF beta and AGR2. I personally recommend the publication of this paper.
I have only a suggestion: the authors could be to add in the figure 3 some images of large part of slides and not only one small section.
Author Response
Reviewer 1:
In this article, the authors investigated the AGR2, an important protein of the cellular machinery responsible for maintaining the epithelial phenotype, which interferes with TGF-β pathway with manifested particularly in cancer cells by the induction of mesenchymal phenotype.
In particular, they performed transcriptomic profiling of a lung cancer cell line with CRISPR-Cas9 mediated AGR2 knockout together with and without TGF-β treatment. The obtained data were subjected to Bioinformatics Microarray Analysis. It was identified significant changes in transcripts linked with focal adhesion and eicosanoid production, in particular arachidonic acid metabolism.
The paper is well organized and well written and represents a contribution to the field of the pathway regulated by TGF beta and AGR2. I personally recommend the publication of this paper.
I have only a suggestion: the authors could be to add in the figure 3 some images of large part of slides and not only one small section.
We would like to comply with this request however the 40x or 20x objectives on our microscope did not provide clear enough slides of the observed structures. The best visualisation was observed with 100x with oil immersion that were used in Figure 3. However, to best address this comment we included additional sections of the cover slides at 100x in Figure S4A and other visualisations at 60x in Figure S4B.
Reviewer 2 Report
In their manuscript, Martisova and collaborator studied changes associated to the downregulation of AGR-2, a molecule that plays an important role in maintaining the epithelial phenotype of cells. They found out that AGR2 plays an important role in focal adhesion and eicosanoid production.
While the authors put a lot of work in this study, some results or their interpretation don’t support other results. The following comments should be addressed to make this manuscript suitable for publication.
Major comments:
- The English needs some improvement throughout the text.
- Figure 4: the results in this figure don’t support what has been found in the previous figures. For most genes selected, one would anticipate that KOAGR2 CTRL would give results similar to scr TGBb. However, in most cases, this is not the case and it looks that KOAGR2 has no effect compared to scr CTR. In the focal adhesion pathway, the only marker that changes as anticipated is VAV3 and in the arachidonic acid metabolism pathway, PTGS2 would fit the model anticipated by the authors where the loss of AGR2 should have an effect similar to a treatment with TGFb. The other markers seem to just change as a response to TGFb stimulation. The authors should screen some of these markers at the protein level. Post transcriptional modifications could have more drastic effects than what the authors observe at the RNA level. In addition, if the authors state that the effect of AGR2 loss is observed mostly when TGFb is added to the cells, they should show on the figures that the difference between scr TGb and KOAGR2 TGFb on one hand and between KOAGR2 CTR and KOAGR2 TGFb on the other hand are significant.
- Migration/invasion should be tested on the 4 models used in Figure 4.
- Generally speaking, using only one cell line to identify pathways involved in EMT and regulated by AGR2 is a major weakness of this manuscript. Figure 4 and Figure 5 could contain more than one cell line to validate the initial observations by RNA seq. That would greatly strengthen this manuscript.
Minor comments:
- Figure 1 B, C: Authors indicated “left” for the graphs showing a downregulation but forgot to mention “right” for those showing an upregulation.
- Line 181: please correct the term “thraction”.
- Line 189: The beginning of the line seems to be out of place. Please correct as needed.
Author Response
Reviewer 2:
In their manuscript, Martisova and collaborator studied changes associated to the downregulation of AGR-2, a molecule that plays an important role in maintaining the epithelial phenotype of cells. They found out that AGR2 plays an important role in focal adhesion and eicosanoid production.
While the authors put a lot of work in this study, some results or their interpretation don’t support other results. The following comments should be addressed to make this manuscript suitable for publication.
Major comments:
The English needs some improvement throughout the text.
This has been addressed, the manuscript was extensively corrected by a native speaker.
Figure 4: the results in this figure don’t support what has been found in the previous figures. For most genes selected, one would anticipate that KOAGR2 CTRL would give results similar to scr TGBb. However, in most cases, this is not the case and it looks that KOAGR2 has no effect compared to scr CTR. In the focal adhesion pathway, the only marker that changes as anticipated is VAV3 and in the arachidonic acid metabolism pathway, PTGS2 would fit the model anticipated by the authors where the loss of AGR2 should have an effect similar to a treatment with TGFb. The other markers seem to just change as a response to TGFb stimulation.
We have mentioned this in the originally submitted manuscript in the discussion section, in revised version see lines 341-346: “However, changes in the level of transcripts used for FA pathway validation were mostly non-significant in AGR2 knockout cells, and were significant only after treatment with TGF-b alone or in combination with KOAGR2. Therefore, the interpretation of our results requires caution and it seems that AGR2 downregulation may act more strongly as a downstream effector of TGF-b in the focal adhesion pathway leading to an enhanced cascade of signaling events favoring cancer progression and dissemination. “
In parallel, we agree that in the results section this should be addressed better and hence we have changed the original statement to: “In AGR2 knockout cells, the mRNA levels behaved similarly to RNAseq data, where the changes in mRNA levels are also lower compared to the effect of TGF-β. “ (Lines 228-230). And we added: “However, these changes were mostly non-significant in RT-qPCR analysis indicating that AGR2 knockout alone is not crucial for FA regulation, but rather shows a synergistic effect in combination with TGF-b.” (230-232)
The following table summarises the above mentioned RNAseq data where in the case of KOAGR2 the fold changes are clearly lower compared to the level of transcripts from TGF-β treated samples.
|
|
scr TGF-β |
KOAGR2 |
KOAGR2 TGF-β |
|
|
COL4A1 |
1,3870 |
0,4466 |
1,7726 |
UP |
|
COL4A2 |
0,9815 |
0,2956 |
1,2739 |
|
|
FLNA |
0,2614 |
0,3836 |
0,6429 |
|
|
VEGFA |
0,7671 |
0,3499 |
1,0396 |
|
|
VINC |
0,2712 |
0,3334 |
0,5248 |
|
|
VAV3 |
-0,2370 |
-0,2352 |
-0,5944 |
DOWN |
|
scr TGF-β |
KOAGR2 |
KOAGR2 TGF-β |
||
|
GPX2 |
-0,9933 |
-0,8948 |
-1,7066 |
DOWN |
|
PLA2G4A |
-0,2035 |
-0,6279 |
-0,8457 |
|
|
PTGS2 |
-0,3968 |
-0,5675 |
-1,1147 |
The authors should screen some of these markers at the protein level. Post transcriptional modifications could have more drastic effects than what the authors observe at the RNA level.
We have screened the following transcripts on protein level using western blot detection: COL4A1, VAV3 and PTGS2 which we included in Figure 4C for A549 and Figure S6C for HT29. This was also added to the manuscript text lines 236-251. The information about used antibodies were added to Materials and Methods section.
In addition, if the authors state that the effect of AGR2 loss is observed mostly when TGFb is added to the cells, they should show on the figures that the difference between scr TGb and KOAGR2 TGFb on one hand and between KOAGR2 CTR and KOAGR2 TGFb on the other hand are significant.
We have added this statistical comparison into the graphs in Fig. 4. Furthermore, we have described the analysis in the results section as follows: “The changes in mRNA levels of focal adhesion associated genes were all significant after TGF-β treatment. In AGR2 knockout cells, the mRNA levels behaved similarly to RNAseq data, where the changes in mRNA levels are also lower compared to the effect of TGF-β. However, these changes were mostly non-significant in RT-qPCR analysis indicating that AGR2 knockout alone is not crucial for FA regulation, but rather shows a synergistic effect in combination with TGF- as shown for COL4A1, COL4A2, VAV3 and VINC (Fig. 4A). Interestingly, AGR2 gene knockout as well as exposure to TGF- led to a significant decrease in in the mRNA levels of all three genes associated with arachidonic acid metabolism (Figure 4B). See lines 227-234
Migration/invasion should be tested on the 4 models used in Figure 4.
We have complied with the reviewers request and performed analysis of invasion using transwell assay on the 4 models shown in Figure 4. Furthermore, we also analysed migration using the wound healing assay (IncuCyte). Both results were included in Figure 3 (B and C, respectively) since it made the best connection with the originally acquired results. The data description was also added to the text at lines 171-175: “Therefore, we also analyzed the invasive and migratory potential of these cells using transwell (Figure 3B) and wound healing assays (Figure 3C), respectively. As expected, scr untreated cells showed the lowest invasion and migration rate. Interestingly, AGR2 knockout seems to play a more prominent role in migratory properties compared to TGF-β treatment in AGR2 expressing cells.” and 346-347. We also added the methods description into the section 4.7.
Generally speaking, using only one cell line to identify pathways involved in EMT and regulated by AGR2 is a major weakness of this manuscript. Figure 4 and Figure 5 could contain more than one cell line to validate the initial observations by RNA seq. That would greatly strengthen this manuscript.
To fulfil this request, we added colorectal cancer cells HT-29 (scr CTR, scr TGF-β, KOAGR2, KOAGR2 TGF-β) as another cell line to validate and generalize our RNAseq data. We tested the mRNA levels of selected genes already used for A549 cells and included these results in Figure S6 A and B. Likewise in the case of A549 we also tested the protein levels of selected transcripts and included them as Figure S6 C. Since Figure 5 already contained HT-29 we left it unchanged. Including HT-29 into independent SI figure seemed to best fit with the rest of the results in our opinion.
Minor comments:
Figure 1 B, C: Authors indicated “left” for the graphs showing a downregulation but forgot to mention “right” for those showing an upregulation.
Line 138 – “right” was added to the figure legend.
Line 181: please correct the term “thraction”.
Corrected to “traction”
Line 189: The beginning of the line seems to be out of place. Please correct as needed.
Unfortunately, we are unable to observe this hence we could not correct that.
Additionally, we also swapped the Biorender figures in our MS and SI for figures suitable for publication with the necessary licences available upon request and mentioned Biorender according to their specifications in the acknowledgement section. Furthermore, we added additional people into the acknowledgements, corrected information on RT-qPCR in section 4.4., clarified treatment regime in section 4.1, and introduced changes which were necessary for the language correction. All of the changes are visible in the submitted revised manuscript through track changes.

Round 2
Reviewer 2 Report
In their manuscript, Martisova and collaborator studied changes associated to the downregulation of AGR-2, a molecule that plays an important role in maintaining the epithelial phenotype of cells. They found out that AGR2 plays an important role in focal adhesion and eicosanoid production.
The manuscript has been substantially improved and provides now conclusions that are in line with their observations.